# Echocardiographic Quantification of Superior Vena Cava (SVC) Flow in Neonates: Pilot Study of Modified Technique

**DOI:** 10.3390/diagnostics12092083

**Published:** 2022-08-28

**Authors:** Benjamim Ficial, Iuri Corsini, Elena Bonafiglia, Enrico Petoello, Alice Iride Flore, Silvia Nogara, Nicola Tsatsaris, Alan M. Groves

**Affiliations:** 1Neonatal Intensive Care Unit, Azienda Ospedaliera Universitaria Integrata Verona, 37126 Verona, Italy; 2Division of Neonatology, Careggi University Hospital of Florence, 50134 Florence, Italy; 3Section of Neurosurgery, Department of Neurosciences, Biomedicine and Movement Sciences, University Hospital, 37126 Verona, Italy; 4Department of Pediatrics, Dell Medical School at the University of Texas, Austin, TX 78723, USA

**Keywords:** neonate, SVC flow, preterm, echocardiography, Phase-contrast MRI, MRI, perfusion, systemic blood flow, cerebral blood flow, doppler

## Abstract

Ultrasound Superior Vena Cava (SVC) flow assessment is a common measure of systemic and cerebral perfusion, although accuracy is limited. The aim of this study was to evaluate whether any improvements in accuracy could be achieved by measuring stroke distance from the instantaneous mean velocity, rather than from peak velocity, and by directly tracing area from images obtained with a high frequency linear probe. Paired phase contrast magnetic resonance imaging (PCMRI) and ultrasound assessments of SVC flow were performed in a pilot cohort of 7 infants. Median postnatal age, corrected gestation and weight at scan were 7 (2–74) days, 34.8 (31.7–37.2) weeks 1870 (970–2660) g. Median interval between PCMRI and ultrasound scans was 0.3 (0.2–0.5) h. The methodology trialed here showed a better agreement with PCMRI (mean bias −8 mL/kg/min, LOA −25–+8 mL/kg/min), compared to both the original method reported by Kluckow et al. (mean bias + 42 mL/kg/min, LOA −53–+137 mL/kg/min), and our own prior adaptation (mean bias + 23 mL/kg/min, LOA −25–+71 mL/kg/min). Ultrasound assessment of SVC flow volume using the modifications described led to enhanced accuracy and decreased variability compared to prior techniques in a small cohort of premature infants.

## 1. Introduction

Superior vena cava (SVC) flow is a widely used echocardiographic measure of cerebral and systemic blood flow in preterm infants which can assess systemic perfusion even in the presence of a PDA [1,2]. SVC flow quantification has produced significant insights into the pathophysiology of low blood flow state during the transitional period in preterm neonates [3] and low SVC flow has been associated with adverse long-term neurological outcomes [4,5].

Ultrasound quantification of flow in any vessel is obtained from the product of heart rate and the stroke volume. The stroke volume is the product of the cross-sectional area, classically obtained from the square of the diameter, assuming the vessel is perfectly circular, and the stroke distance or velocity time integral (VTI), a measure of how far the blood travels during a single cardiac cycle [6].

In the initial description of SVC flow quantification, as described by Kluckow et al., the diameter was measured from a parasternal long axis view, at the entrance of the vein into the right atrium. The diameter is then squared to calculate SVC cross-sectional area. VTI is obtained by measuring SVC blood flow velocity from a subcostal view, placing the Pulse Wave Doppler (PWD) gate at the entrance of the vessel into the right atrium [1].

However, limited inter and intraobserver repeatability of this original approach of assessment of SVC flow was reported [7,8]. For this reason, consensus statements still recommend caution when interpreting SVC flow measurements at the cotside [9].

In order to optimize SVC flow quantification we previously suggested directly tracing SVC cross-sectional area from a short axis view at the level of the right pulmonary artery (RPA), rather than calculating the cross-sectional area from the diameter, since the vessel is not perfectly circular [10]. Moreover, Harabor et al. proposed to measure SVC VTI from a suprasternal rather than a subcostal view, to avoid abdominal motion due to respiration [11].

Finally, we combined the last two: we assessed SVC flow velocity from a suprasternal view, and we measured SVC area from a short axis view, performing both measures at the level of the RPA. This modified approach showed significantly improved repeatability compared to the traditional approach. However, the accuracy was still suboptimal, when compared against phase contrast magnetic resonance imaging (PCMRI) [12].

In fact, suprasternal VTI systematically overestimated the stroke distance measured by PCMRI. We hypothesized that echo provides an increased estimate of flow velocity due to the presence of non-laminar flow in the SVC. VTI is generally obtained by integrating the peak velocity over time. In the presence of non-laminar flow, the assumption that all flow is at the peak velocity leads to an overestimation of the true blood velocity. Further improvements in accuracy may be achieved by measuring VTI from the instantaneous mean, rather than peak velocity. Moreover, SVC axial area underestimated vessel area when compared to PCMRI. We hypothesized that further improvements in accuracy of echo measurements of SVC cross-sectional area may be achieved using a high frequency linear probe, to enhance vessel edge definition.

The aim of this study was to evaluate whether any improvements in accuracy of SVC flow quantification could be achieved by measuring VTI from the instantaneous mean velocity and by measuring area from images obtained with a high frequency linear probe. Accuracy of the echocardiographic SVC flow measurements was compared against gold standard PCMRI measures.

## 2. Methods

This was a prospective, observational study. Infants undergoing paired PCMRI and echo assessments, were recruited from stable infants admitted to the neonatal units or postnatal wards at Queen Charlotte’s and Chelsea Hospital, London, UK, from June to November 2012.

Exclusion criteria were congenital heart diseases (CHD), including bilateral SVC and persistent pulmonary hypertension of the newborn (PPHN), except for a PDA or PFO. All admitted infants meeting the eligibility criteria within the study period were enrolled (convenience sampling). No formal sample size calculation was done. Echo and PCMRI scans were carried out with ethics committee approval and signed parental consent. Attending clinicians were blind to SVC flow findings.

### 2.1. PCMRI Acquisition

PCMRI was performed on a 3T scanner (Philips, Best, Netherlands) during natural sleep after a feed, without using sedation. The methodology has previously been described and is summarized below [13,14].

PCMRI was performed on a 3T scanner (Philips, Best, Netherlands), available at both sites (Queen Charlotte’s and Chelsea Hospital and St. Thomas’ Hospital). A specialized 8 channel pediatric body receive coil for infants > 2 kg, or a small extremity receive coil for infants < 2 kg were used. Infants were allowed to fall into a natural sleep after a feed, without using sedation. They had continuous monitoring of heart rate, oxygen saturation and temperature. They received nasal CPAP or low flow oxygen support as clinically indicated and a specially trained pediatrician was in attendance at all times. Pilot scans were obtained to identify position of the SVC and pulmonary arteries. Single slice PCMRI was obtained at the level of the RPA with the imaging plane perpendicular to the centreline of the vessel, so as to minimize partial volume effect. Acquisition sequences had spatial resolution of 0.6 mm in-plane, slice thickness 4 mm and TR/TE = 5.9/3.1 ms, 3 signal averages and 20 phases per cardiac cycle. Field of view and matrix were altered to maintain spatial resolution at 0.6 mm while minimizing scan duration. The velocity encoding was calibrated for the range of ±60 cm/s. Depending on heart rate and heart rate stability the acquisition time for each 2D PCMRI scan ranged between 2 and 4 min. No respiratory compensation techniques were employed. Flow volume quantification for phase contrast datasets was performed using a commercial workstation (Philips ViewForum). Automated vessel edge detection was used with manual correction where necessary. Once defined in the first cardiac phase the software tracks the vessel of interest over the cardiac cycle using edge detection algorithms. Flow is then calculated at each time point of the cardiac cycle, generating a flow curve and volume of flow value for the vessel of interest. SVC flow quantification was performed offline by one observer blinded to echo results.

### 2.2. Echocardiographic Measures

All echocardiograms were performed by a single investigator (B.F.), with infants asleep or quietly awake. No sedation was used. A Vivid 7 ultrasound machine (GE Healthcare, Milwaukee, WI, USA) was used. Echocardiography was performed as soon as possible after the PCMRI, while infants were still asleep or quietly awake. Echocardiographic measures of SVC flow according to all three techniques were obtained as described below. In all instances SVC diameter and cross-sectional area were obtained from 3–5 consecutive cycles. In all instances SVC flow velocity was obtained from 8–10 consecutive cycles, to reduce the impact of respiratory variability. A simplified structural echocardiogram was performed to exclude significant CHD. All flow quantification was performed offline by a single investigator (B.F.) blinded to the PCMRI results.

Characteristics of the three echocardiographic methods of SVC flow assessment are detailed below and summarized in Table 1.

#### 2.2.1. Method 1—Traditional Superior Vena Cava Flow Technique

SVC diameter was assessed from a modified parasternal long axis view as described by Kluckow et al. [1]. High-definition zoom was used to focus on the SVC as it opens into the right atrium (Figure 1A). Maximum and minimum diameters through the cardiac cycle were measured offline from B mode images.

SVC flow velocity was assessed from a subcostal view as described by Kluckow et al. [1]. The PWD gate was placed at the SVC-right atrium junction (Figure 1B). SVC VTI was calculated offline from SVC peak velocity.

#### 2.2.2. Method 2—Modified Superior Vena Cava Flow Technique

SVC area was assessed directly from an axial/short axis view, at the level of the RPA, using high-definition zoom, with a 10 MHz sector probe. Maximum and minimum cross-sectional areas through the cardiac cycle were measured offline from the B mode images (Figure 2A).

SVC flow velocity was assessed from a suprasternal view following the SVC from its origin at the innominate vein confluence to the right atrium. Angle correction of flow velocity was not used. The PWD gate was placed at the level of the RPA. VTI was calculated offline from SVC peak velocity (Figure 3).

#### 2.2.3. Method 3—Modified Superior Vena Cava Flow Technique

SVC area was assessed directly from an axial/short axis view, at the level of the RPA as in Method 2, but in this case using a high frequency linear probe (12 MHz). Maximum and minimum area through the cardiac cycle were measured offline from the B mode images (Figure 2B).

The ultrasound imaging acquisition procedure is detailed in Figure 4.

SVC flow velocity was assessed from a suprasternal view as in Method 2, but in this case VTI was calculated from the instantaneous mean velocity rather than the instantaneous peak velocity. Measures were made off-line with Tomtec Arena, version TTA2 41.00, (Tomtec, Unterschleißheim, Germany) (Figure 3).

The ultrasound imaging acquisition procedure is detailed in Figure 5.

### 2.3. Statistical Analysis

Data are expressed as median [range] for non-parametric variables, mean and standard deviation (SD) for parametric variables and percentage for categorical variables. Agreement between SVC measurements according to the three techniques by ultrasound and PCMRI was assessed using: Bland-Altman analysis. Mean bias, limits of agreement (LOA) were calculated. Repeatability index (RI = 1.96 × SD of differences between measures/mean of measures) was also calculated to allow comparison of repeatability [15]. Coefficient of variation (COV) was calculated as follows: (standard deviation of differences between repeated measurements/arithmetic mean of all repeated measurements) × 100%. *p* values < 0.05 were considered significant. Data were analyzed with SPSS 20 (SPSS, Chicago, IL, USA).

## 3. Results

Paired PCMRI and ultrasound assessments of SVC flow according to the three techniques were performed in a pilot cohort of 7 infants. Clinical characteristics of the enrolled neonates are detailed in Table 2. Median [range] gestation and weight at birth were: 32.5 [24.7–37] weeks and 1625 [640–2688] g respectively. Median postnatal age at scan was 7 [2–74] days, with corrected gestation 34.8 [31.7–37.2] weeks and weight at scan 1870 [970–2660] g.

Median interval between PCMRI and ultrasound scans was 0.3 [0.2–0.5] hours. No infant had a significant change in clinical condition, medications, feeding and sleep state, nor received a blood transfusion, vasoactive medication, diuretics, or treatment for a PDA between the scans. Three neonates were receiving nasal continuous positive airway pressure support during both MRI and ultrasound scans. One neonate had a PDA. No infant had PPHN or CHD. No infant had death, bronchopulmonary dysplasia, severe intraventricular hemorrhage, or necrotizing enterocolitis before discharge.

SVC flow, SVC area, VTI and heart rate assessed by PCMRI, and the three ultrasound techniques are shown in Table 3.

### 3.1. Method 1

Compared to PCMRI, SVC flow quantified by ultrasound Method 1 showed a mean bias of +42 mL/kg/min, LOA −53–+137 mL/kg/min. Individually SVC area calculated from diameter and VTI measured subcostally showed: mean bias −11 mm^2^, LOA −23–+0.4 mm^2^ and mean bias +11.1 cm, LOA +4.9–+17.3 cm, respectively (Table 4).

### 3.2. Method 2

Compared to PCMRI, SVC flow quantified by ultrasound Method 2 showed a mean bias of +23 mL/kg/min, LOA −25–+71 mL/kg/min. SVC area directly assessed axially and VTI measured suprasternally showed: mean bias −8 mm^2^, LOA −15–−0.5 mm^2^ and mean bias +6 cm, LOA +0.7–+11 cm, respectively (Table 4).

### 3.3. Method 3

Compared to PCMRI, SVC flow quantified by ultrasound Method 3 showed a mean bias of −8 mL/kg/min, LOA −25–+8 mL/kg/min. SVC area directly assessed axially using a high frequency linear probe and VTI measured suprasternally using instantaneous mean velocity showed: mean bias −2.7 mm^2^, LOA −10–+4.8 mm^2^ and mean bias +0.5 cm, LOA −1–+2.1 cm, respectively (Table 4).

## 4. Discussion

The main finding of the current study is that directly tracing SVC area from a short axis view with a linear high frequency probe and calculating suprasternal VTI from mean rather than peak velocity improved the accuracy of ultrasonic SVC flow quantification in this small pilot study.

The methodology trialed here (‘Method 3’) showed a better agreement with PCMRI compared to both the original method reported by Kluckow et al. (‘Method 1’), and our own prior adaptation (‘Method 2’).

There is an urgent need to improve hemodynamic monitoring in preterm neonates since the clinical parameters traditionally used to assess the cardiovascular adequacy are inadequate to identify low blood flow state [16]. In fact, arterial blood pressure, despite its widespread use, poorly reflects cerebral and systemic perfusion, since it is also dependent on peripheral vascular resistance. The latter has a marked variability in neonates and cannot be directly quantified [17,18]. To confirm the poor reliability of using blood pressure alone to assess the cardiovascular wellbeing, to the best of our knowledge, there is no data demonstrating any improvement in outcome with the treatment of low blood pressure [19]. Moreover, other clinical parameters (e.g., heart rate, lactate, renal function, capillary refill time, etc.) showed a low ability to detect low blood flow state [16]. Point of care ultrasound (POCUS) is increasingly used to assess hemodynamic status on the NICU, and a reliable marker of cerebral and systemic hypoperfusion would be of huge value, allowing practitioners to individualize therapy and improve clinical outcomes [20].

However, multiple studies have suggested that quantification of SVC flow is less accurate than other standard hemodynamic measures in the newborn [7,8,12], and current guidelines of neonatal functional ultrasound either do not mention the technique or recommend caution when using it [20,21].

Despite concerns over its repeatability, quantification of SVC flow is a relatively widely-used ultrasonic marker of systemic blood flow in preterm infants. Data from national and international surveys has shown that around 40% of neonatologists who use POCUS include assessment of SVC flow in their hemodynamic assessments [22,23].

In 2017 we proposed two modifications to the original approach:

1—visualize and quantify SVC area directly from the axial rather than sagittal view. This allows the practitioner to avoid multiplying errors when squaring vessel radius to estimate area, and allows a repeatable anatomic landmark for measuring size of a vessel which is gradually increasing as it ‘opens up’ into the right atrium

2—visualize and quantify SVC flow from the suprasternal view. This allows the practitioner to have the ultrasound probe closer to the vessel being interrogated, and to reduce the impact of respiratory motion on placement of the pulsed wave Doppler gate.

In our opinion, these were important steps towards the optimization of SVC flow quantification [12]. The improved repeatability we demonstrated was subsequently reproduced by Bischoff et al. [24].

However, despite improved accuracy and repeatability the 2017 method continued to systematically overestimate stroke distance and underestimate SVC area. We therefore now examined two additional modifications to the original approach:

3—when visualizing the SVC area axially use a high frequency linear probe to improve edge delineation. The parallel rather than diverging path of ultrasound beams from linear as opposed to sector ultrasound probes has been shown to improve visualization of other vessels [25] and showed superiority to sector probes in phantom models [26]. Our initial results suggest that improved image resolution may result in better accuracy of SVC area measurements, with mean bias from PCMRI decreasing from −8 mm^2^ to −2.7 mm^2^. However, it is noted that this modification had minimal impact on limits of agreement.

4—when quantifying SVC flow velocity trace the velocity time integral from the instantaneous mean velocity rather than the instantaneous peak velocity. In the presence of non-laminar flow in a vessel, the assumption that all flow is at peak velocity will always lead to an overestimation of the true blood velocity. This effect may be considered marginal in the presence of arterial flow, where the range of velocities is narrow and most of the blood cells are travelling at the same velocity. Instead, in the presence of turbulent flow, both preclinical and clinical studies showed that PWD overestimates VTI. In fact, in case of turbulent flow, the blood cells cross the sample volume of the PWD at different velocities in random and the signals backscattered by blood cells will increase the Doppler spectral broadening [27,28,29,30].

Moreover, the degree of the Doppler spectral broadening may be further increased by an artifact related to the actual mechanism of delivering and receiving acoustic energy, the so-called intrinsic spectral broadening, that ultimately leads to a further overestimation of the true peak velocity [29].

Our initial results suggest that utilizing mean rather than peak velocity over time to calculate VTI, resulted in improved accuracy of SVC VTI, with mean bias from MRI decreasing from 6 cm to 0.5 cm. Limits of agreement also narrowed significantly from ±5.8 cm to ±1.5 cm.

The combination of both modifications significantly improved accuracy of quantification of total SVC flow, with mean bias decreasing from 23 mL/kg/min to −8 mL/kg/min. Limits of agreement also narrowed significantly from ±48 mL/kg/min to ±17 mL/kg/min. The accuracy of the improved approach of SVC flow quantification is now similar to that of left ventricular output (LVO), a common measure of systemic blood flow, as we previously reported [10].

In our opinion these modifications may represent a further step towards the use of ultrasound SVC flow quantification in regular clinical practice. However, we acknowledged that further validation is needed, and our results should be confirmed by other groups, reproducibility should be assessed, normative ranges should be defined and associations to relevant clinical outcomes should be evaluated.

This study has several limitations. First, the number of infants with paired echo and PCMRI measurements is small, and no formal sample size calculation was done. However, to the best of our knowledge, these are the only data comparing echo assessment of SVC flow according to multiple techniques in neonates. Study of large cohorts of preterm neonates with paired ultrasound and MRI scans is extremely challenging and may not be feasible without delaying treatment or causing any potential discomfort to the neonate. Second, the validation of SVC flow quantification was performed in relatively stable infants who were older and more mature than many premature infants at risk of hypoperfusion. It is possible that reliability of the technique may be lower in smaller sicker infants. Though we have no reason to believe that the relative improvement in reliability of the modifications described would not still be present relative to prior approaches. Third, PCMRI is not unanimously considered a gold standard for the assessment of blood flow in neonates, since suboptimal temporal and spatial resolution may result in errors in quantification of blood flow [31]. However, PCMRI is widely used to assess hemodynamics in adults [32]. Acknowledging that invasive techniques (such as the Fick method and thermodilution) may not be feasible in this patient population, in our opinion PCMRI is the best comparator currently available for the validation of echocardiographic findings. Groves et al. previously validated PCMRI sequences against a gold standard ex-vivo flow phantom and then against MRI volumes of cardiac output generated from end-diastolic and end-systolic left ventricular volumes showing that PCMRI has extremely high accuracy and reproducibility in newborns [13]. Moreover, they internally validated PCMRI sequences of systemic perfusion by comparing LVO to the sum of SVC and descending aortic flow in neonates without PDA. Lastly, they demonstrated the low scan-rescan variability of SVC flow by PCMRI [14].

## 5. Conclusions

Ultrasound assessment of SVC flow volume using the modifications described led to enhanced accuracy and decreased variability compared to prior techniques in a small cohort of premature infants. However, we discourage practitioners utilizing POCUS to assess SVC flow from applying this modified method in daily clinical practice. In fact, further validation is needed, reproducibility should be assessed, association with relevant clinical outcomes should be confirmed and normal reference values should be developed. We wish that this pilot data prompt further validation studies, with the potential to demonstrate that measures of SVC flow volume can now be made with the same accuracy as other accepted measures of hemodynamic status in the newborn [10].

## Figures and Tables

**Figure 1 diagnostics-12-02083-f001:**
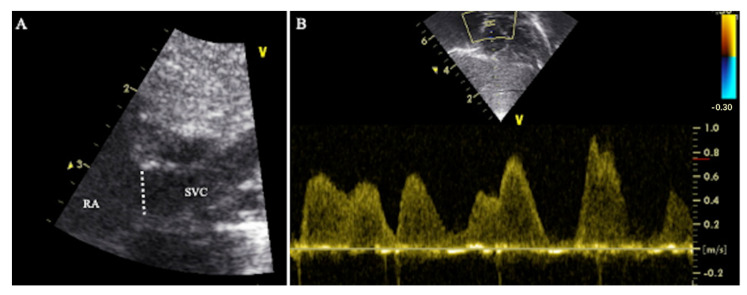
Traditional technique of SVC flow assessment (Method 1). (**A**) High-definition zoom on the SVC-right atrium junction from a modified parasternal long axis view. The dotted line represents the SVC diameter. (**B**) Doppler profile of blood flow velocity in the SVC. The sample gate of pulsed wave Doppler was placed at the SVC-right atrium junction from a subcostal view. RA = right atrium, SVC = superior vena cava.

**Figure 2 diagnostics-12-02083-f002:**
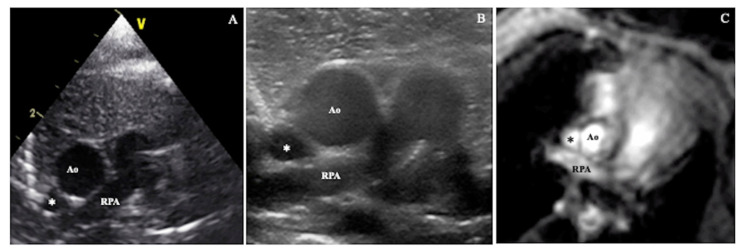
SVC cross-sectional area assessed by: (**A**) sector probe according to Method 2, (**B**) high frequency linear probe according to Method 3, (**C**) phase contrast magnetic resonance imaging. Ao = aorta, RPA = right pulmonary artery, * = superior vena cava.

**Figure 3 diagnostics-12-02083-f003:**
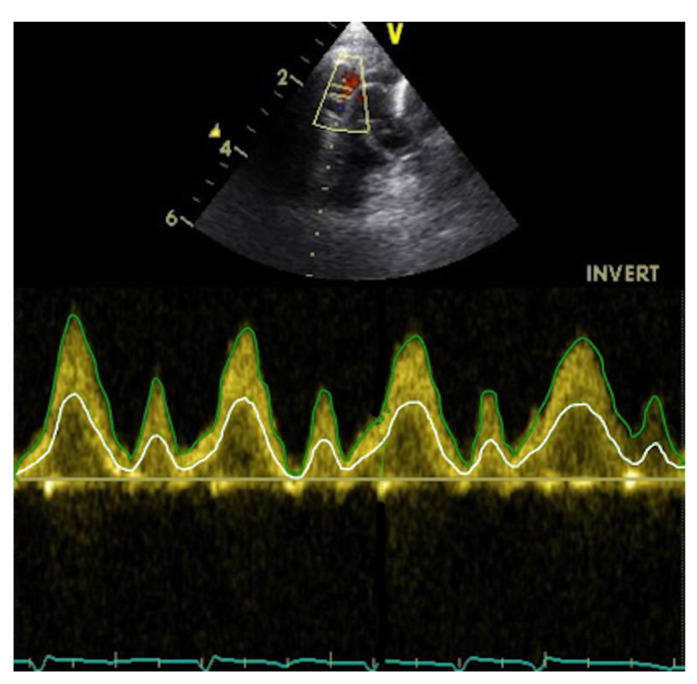
Spectral Doppler profile of blood flow velocity in the SVC, assessed from a suprasternal view. The green line represents the peak blood flow velocity, the white line the mean blood flow velocity.

**Figure 4 diagnostics-12-02083-f004:**
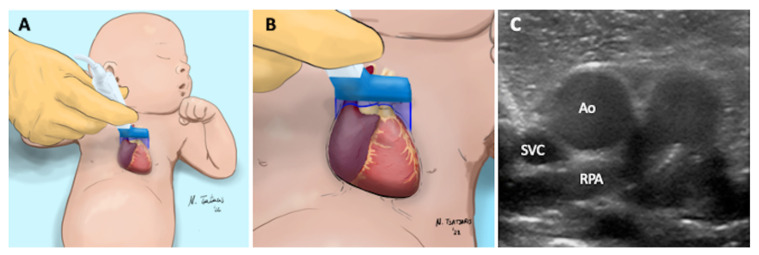
Axial or short axis view of the SVC. (**A**) The high frequency linear probe is placed near the left sternal edge (second intercostal space). (**B**) The ultrasound beam is perpendicular to SVC at the level of the right pulmonary artery. (**C**) The following structures are visualized: superior vena cava (SVC), aorta (Ao), right pulmonary artery (RPA).

**Figure 5 diagnostics-12-02083-f005:**
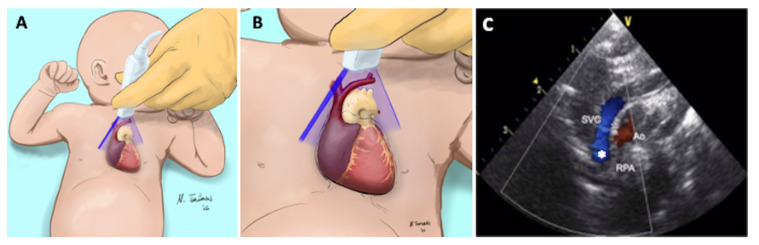
Suprasternal view of the SVC. (**A**) The sector probe is placed near the suprasternal notch. The orientation marker is pointed towards 1:00 to 2:00. (**B**) The ultrasound beam is parallel to the SVC, (**C**) The following structures are visualized: superior vena cava (SVC), aorta (Ao), right pulmonary artery (RPA). The sample volume of the pulsed wave doppler (*) is placed in the SVC at the level of the RPA.

**Table 1 diagnostics-12-02083-t001:** Characteristics of the three methods of SVC flow assessment.

	Method 1	Method 2	Method 3
Equipment:			
- VTI	Sector probe	Sector probe	Sector probe
- SVC CSA	Sector probe	Sector probe	High frequency linear probe
Echo views:			
- VTI	Subcostal	Suprasternal	Suprasternal
- SVC CSA	Long axis parasternal	Short axis parasternal	Short axis parasternal
Off-line analysis:			
- VTI	From peak velocity	From peak velocity	From mean velocity
- SVC CSA	Calculated from diameter	Directly traced	Directly traced

SVC = superior vena cava; VTI = velocity time integral; CSA = cross-sectional area.

**Table 2 diagnostics-12-02083-t002:** Clinical characteristics of the cohort of neonates.

	Cohort (*N* = 7)
Gestation (wk)	32.5 [24.7–37]
Birth weight (g)	1625 [640–2688]
Female	2 (28.5%)
Cesarean delivery	4 (57.1%)
5-Min Apgar score	9 [7–10]
Postnatal age at scan (days)	7 [2–74]
Corrected gestation (wk) at scan	34.8 [31.7–37.2]
Weight (g) at scan	1870 [970–2660]
Non-invasive respiratory support	3 (42.8%)
No respiratory support	4 (57.1%)

Values are presented as median [range], and count (%). GA = Gestational age.

**Table 3 diagnostics-12-02083-t003:** Mean (SD) of SVC diameter, SVC area, VTI, HR and SVC flow assessed by ultrasound according to the three methods and PCMRI.

	SVC Flow Assessment by Echocardiography			SVC Flow Assessment by PCMRI
	Method 1	Method 2	Method 3	
SVC diameter (mm)	4.1 (0.7)	NA	NA	NA
SVC cross sectional area (mm^2^)	11 (2)	15 (4)	20 (4)	23 (6)
SVC VTI or stroke distance (cm)	17.7 (3.2)	12.6 (2.9)	7.1 (1.5)	6.6 (1.4)
HR (bpm)	143 (10)	139 (9)	139 (9)	146 (12)
SVC flow (ml/kg/min)	174 (54)	153 (50)	121(39)	130 (38)

SVC = superior vena cava; PCMRI = phase contrast magnetic resonance imaging; NA = not applicable; VTI = velocity time integral; HR = heart rate.

**Table 4 diagnostics-12-02083-t004:** Agreement between PCMRI and the three echocardiographic techniques of SVC flow assessment.

	Bias	95% LOA	RI	COV
Method 1				
SVC flow	+42 mL/kg/min	−53/+137 mL/kg/min	63%	32%
Area	−11 mm^2^	−23/+0.4 mm^2^	69%	35%
VTI	11.1 cm	4.9/17.3 cm	51%	26%
Method 2				
SVC flow	+23 mL/kg/min	−25/+71 mL/kg/min	48%	17%
Area	−8 mm^2^	−15/−0.5 mm^2^	40%	20%
VTI	6 cm	0.7/11 cm	55%	28%
Method 3				
SVC flow	−8 mL/kg/min	−25/+8 mL/kg/min	13%	7%
Area	−2.7 mm^2^	−10/+4.8 mm^2^	35%	17%
Suprasternal VTI	0.5 cm	−1/+2.1 cm	23%	11%

SVC = superior vena cava; PCMRI = phase contrast magnetic resonance imaging; VTI = velocity time integral; LOA = limits of agreement; COV = coefficient of variation; RI = repeatability index.

## Data Availability

Data are available upon reasonable request.

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
