# Peer review of "Echocardiographic Quantification of Superior Vena Cava (SVC) Flow in Neonates: Pilot Study of Modified Technique"

_diagnostics, 2022, doi:10.3390/diagnostics12092083_

Round 1
Reviewer 1 Report
Thank you for the modified version of the technique to accurately determine SVC flow in premature infants.
I have some practical concerns:
1. typically, daily weight, heart rate, basic renal profile test, and lactate will provide a great assessment of cardiac output. Does this fancy method is necessary for the point of care
2. Authors ruled out CHD, but did they rule out PFO and PPHN in premature babies in the first month of life?
3. Diastolic function of ventricles both left and right in premature infants are abnormal and there is a process of maturation of diastolic function? How does this affect the SVC flow in preterm infants?
4. What is the status of the lungs, are these babies on diuretics?
5. Authors used Bald-Altman analysis between 3 methods, but all the data are collected and measured by one investigator. Can authors do this with another investigator to find out inter-observed variability and reproducibility?
Author Response
Response to Reviewer 1 Comments
Point 1. Thank you for the modified version of the technique to accurately determine SVC flow in premature infants.
Response 1: We thank the reviewer for their positive comment.
Point 2. I have some practical concerns:
- typically, daily weight, heart rate, basic renal profile test, and lactate will provide a great assessment of cardiac output. Does this fancy method is necessary for the point of care.
Response 2: We thank the reviewer for raising these important points. Cardiac output or systemic perfusion is traditionally assessed clinically, using a combination of various parameters, as the reviewer mentioned (lactate, heart rate, blood pressure, renal function, etc.). However, in neonates, there is a large body of evidence that, despite their widespread use, these clinical parameters are inadequate to identify low blood flow state. In fact, these parameters have many confounding factors and are poorly related to perfusion. For instance, blood pressure does not equate to perfusion, that is dependent also on vascular resistance. The latter cannot be directly quantified and has a marked variability in neonates, in particular if preterm. To confirm the poor reliability of blood pressure as a measure to assess cardiovascular adequacy, there is no data demonstrating an improvement in outcome with the treatment of low blood pressure. Moreover, neonatal creatinine in the first days of life reflects maternal rather than neonatal renal function, lactate has a low sensibility to identify low blood flow state, a change in daily weight has a low accuracy and depends on many factors other than cardiac output. Therefore, there is an urgent need for reliable markers of systemic perfusion due to the poor association between clinical parameters and cardiac output in neonates. We have acknowledged that this modified method to quantify SVC flow should not be applied at the cotside before further validation and development of normal reference ranges, However, we feel that this technique is promising. In our hands, the Method 3 offered an improved agreement with PCMRI, and, while waiting for feedback from other researchers, we are still hopeful that our modifications may take the technique closer to being usable in regular clinical practice with improved reliability. We added discussion on such points on lines (289-298, 397-403).
Point 3. 2. Authors ruled out CHD, but did they rule out PFO and PPHN in premature babies in the first month of life?
Response 3: We thank the reviewer for this observation. We ruled out PPHN (exclusion criteria). However, we included neonates with PFO. We clarified that in the method section (Lines 85-86) and in the Results sections (Lines 221-222).
Point 4. 3. Diastolic function of ventricles both left and right in premature infants are abnormal and there is a process of maturation of diastolic function? How does this affect the SVC flow in preterm infants?
Response 4: We thank the reviewer for raising this important point. We do agree with the reviewer that the diastolic function improves in preterm infants during the first days of life and that this may have an impact on SVC flow. Unfortunately, this is beyond the scope of the current study. A prospective observational study with serial assessments of SVC flow and diastolic function at various time points is needed to explore this interesting hypothesis.
- What is the status of the lungs, are these babies on diuretics?
Response 5: We thank the reviewer for this observation. We added a table to better clarify the type of respiratory support (Table 2, Line 213) and we specified that no infant received diuretics at the time of scan (Line 219).
- Authors used Bald-Altman analysis between 3 methods, but all the data are collected and measured by one investigator. Can authors do this with another investigator to find out inter-observed variability and reproducibility?
Response 6: We thank the reviewer for this interesting observation. Bland-Altman analysis is commonly used to assess intra- and inter-observer reproducibility of repeated measurements, according to the same technique. Moreover, it is the method of choice to compare measurements obtained by two different diagnostic techniques in the absence of a true gold standard. Therefore, given the fact that PCMRI is not universally recognized as the gold standard to assess cardiac output in neonate, Bland-Altman analysis was the method of choice to compare the three ultrasound methods in terms of agreement with PCMRI. Our study protocol did not allow for repeated measurements of SVC flow to assess scan rescan intra- and inter-observer reproducibility and the number of patients is too small to assess intra- and inter-observer reproducibility of the off-line analysis, compared to similar studies. Further studies are needed to assess reproducibility. We acknowledged this on lines 397-399.

Reviewer 2 Report
This is an original article describing a new method of measuring SVC flow. I have several comments here:
1. This study using PCMRI as a gold standard. However, infants’ cardiac output changes very quickly. It may need more references to persuade readers that it can be used as the gold standard.
2. From table 3, Kluckow’s method (Method 1) has the best RI. Does it mean that it has a better reliability?
3. From table 3, Method 3 has the worst RI. It might mean that it has poor reliability. Can the authors explain it more?
4. Although case numbers are very limited, there should be a table for demographic data.
5. This is an article describing a new method to measure SVC flow. Detail figures of measurement should be prepared in a step-by-step method.
6. Is there any poor outcome happened for those 7 enrolled patients?
Author Response
Response to Reviewer 2 Comments
Point 1. This is an original article describing a new method of measuring SVC flow. I have several comments here:
- This study using PCMRI as a gold standard. However, infants’ cardiac output changes very quickly. It may need more references to persuade readers that it can be used as the gold standard.
Response 1: We thank the reviewer for this observation. PCMRI is a widely used clinical application to assess hemodynamics in adults and it has been proven to be a robust and reliable tool to measure flow independently of the anatomic localization. We acknowledge that PCMRI is not unanimously considered a gold standard in neonates. However, given the fact that invasive methods (such as the Fick method and thermodilution) are not applicable in neonates, we feel that PCMRI is the best comparator currently available for the validation of echocardiographic measures of cardiac output. The research team lead by dr. Alan Groves, that I joint for this research project on systemic blood flow measurements in neonates, had previously validated these PCMRI sequences against a gold standard ex-vivo flow phantom and then against MRI volumes of cardiac output generated from end-diastolic and end-systolic left ventricular volumes in preterm infants, showing that PCMRI can be successfully applied to assess blood flow in neonates. Moreover, we have internally validated PCMRI sequences of systemic perfusion by comparing LVO to the sum of SVC and descending aortic (DAo) flow in neonates without a ductus arteriosus (repeatability index-13.2%). Lastly, we have demonstrated the low scan-rescan variability of SVC flow by PCMRI (repeatability index-12.8%). We modified the text following the reviewer’s suggestion (390-393).
Point 2. From table 3, Kluckow’s method (Method 1) has the best RI. Does it mean that it has a better reliability?
Response 2: We thank the reviewer for this question. Repeatability Index (RI) is equal to 1.96 x SD of differences between measures / mean of measures. Therefore, RI is inversely related to reproducibility. The highest RI (Method 1), has the worst repeatability. We added more details on RI (Lines 200-202).
Point 3. From table 3, Method 3 has the worst RI. It might mean that it has poor reliability. Can the authors explain it more?
Response 3: See above. The lowest RI (Method 3), has the better repeatability. We added more details on RI (Lines 200-202).
Point 4. Although case numbers are very limited, there should be a table for demographic data.
Response 4: We thank the reviewer for this observation. We have added a table for demographic data (Table 2, Line 213)
Point 5. This is an article describing a new method to measure SVC flow. Detail figures of measurement should be prepared in a step-by-step method.
Response 5: We thank the reviewer for this observation. We have added Figure 4 (Lines 172-176) and 5 (Lines 189-194), following the reviewer’s suggestion.
Point 6. Is there any poor outcome happened for those 7 enrolled patients?
Response 6: Thanks for pointing this out. No infant had death, bronchopulmonary dysplasia, severe intraventricular hemorrhage or necrotizing enterocolitis before discharge. (Lines 221-223)

Reviewer 3 Report
Ficial and colleagues report on a small cohort of neonates in whom they attempt to evaluate and improve the accuracy of ultrasound measurement of superior vena cava flow, by comparison with phase-contrast MRI. The study design is simple, but the measurements are technically challenging.
The manuscript contains an appropriate amount of technical information to describe and illustrate the methods used, and the data analysis and discussion are adequate; it is generally well written, though it will benefit from some corrections and clarifications, as detailed below.
The Abstract accurately summarizes the contents of the paper.
The Introduction is succinct and it contains adequate references to relevant literature; it is written clearly, in the absence of a specific hypothesis is not problematic in this context.
The Methods include a detailed research protocol on the methods of measurement of analysis. However, there is no information on an a priori power analysis, and it is unclear how the researchers decided the study should be stopped; it is also unclear if the analyses were all conducted at the end of the study or also on interim data.
In the Results section, both Table 2 and Table 3 are well designed and useful to communicate the study’s findings. However, it is not necessary nor desirable to repeat all the same numerical information from the tables in the body text. I suggest that the authors selectively minimize repeating the numbers provided under each method, to facilitate rapid comprehension by readers; it would be ideal if table 3 and abbreviated sections 3.1-3.3 were on the same physical page, although this may be difficult to achieve with the present layout.
The Discussion is balanced and presented clearly. The authors acknowledge the various limitations of the study, including the small number of preterm infants, who were also stable so that the variability of measurements reported here may not be generalized able to sick neonates, in whom these measurements may be potentially most useful. Furthermore, all the echocardiograms were performed by a single researcher, and additional variability may be introduced by using multiple echocardiographers, as one would expect in the clinical arena. The authors appropriately call for further validation of their findings, in both the discussion and conclusions.
The manuscript contains multiple minor grammatical errors or slightly awkward wording which will be readily apparent to native English speakers (e.g., line 40, “begins to opens up…”). Other lines where careful reading will reveal minor grammatical flaws include 146, 204, 251, 266, 280, 292, in lines 262-264. Finally, it is unclear whether the format of the references is that used by MDPI/Diagnostics.
Author Response
Response to Reviewer 3 Comments
Point 1. Ficial and colleagues report on a small cohort of neonates in whom they attempt to evaluate and improve the accuracy of ultrasound measurement of superior vena cava flow, by comparison with phase-contrast MRI. The study design is simple, but the measurements are technically challenging. The manuscript contains an appropriate amount of technical information to describe and illustrate the methods used, and the data analysis and discussion are adequate; it is generally well written, though it will benefit from some corrections and clarifications, as detailed below.
Response 1: We thank the reviewer for their positive comment and for putting our work into context.
Point 2. The Abstract accurately summarizes the contents of the paper.
Response 2: We thank the reviewer for their positive comment.
Point 3. The Introduction is succinct and it contains adequate references to relevant literature; it is written clearly, in the absence of a specific hypothesis is not problematic in this context.
Response 3: We thank the reviewer for their positive comment.
Point 4. The Methods include a detailed research protocol on the methods of measurement of analysis. However, there is no information on an a priori power analysis, and it is unclear how the researchers decided the study should be stopped; it is also unclear if the analyses were all conducted at the end of the study or also on interim data.
Response 4: Thank you for pointing this out. Following your suggestion, we have clarified the design of the study. All admitted infants meeting the eligibility criteria within the study period were enrolled (convenience sampling). No formal sample size calculation was done (Lines 87-88). We agree that this is a potential limitation of the study and we have added this as a limitation (Lines 363-364). We would stress that coordinating MRI and echo scans in preterm neonates may not always be feasible without delaying treatment or causing any potential discomfort to the baby. Finally, we did not do analysis on interim data, but we conducted the analysis after the recruitment ended.
Point 5. In the Results section, both Table 2 and Table 3 are well designed and useful to communicate the study’s findings. However, it is not necessary nor desirable to repeat all the same numerical information from the tables in the body text. I suggest that the authors selectively minimize repeating the numbers provided under each method, to facilitate rapid comprehension by readers; it would be ideal if table 3 and abbreviated sections 3.1-3.3 were on the same physical page, although this may be difficult to achieve with the present layout.
Response 5: We thank the reviewer for this observation. We modified the text following the reviewer’s suggestions (Lines 230-245).
Point 6. The Discussion is balanced and presented clearly. The authors acknowledge the various limitations of the study, including the small number of preterm infants, who were also stable so that the variability of measurements reported here may not be generalized able to sick neonates, in whom these measurements may be potentially most useful. Furthermore, all the echocardiograms were performed by a single researcher, and additional variability may be introduced by using multiple echocardiographers, as one would expect in the clinical arena. The authors appropriately call for further validation of their findings, in both the discussion and conclusions.
Response 6: We thank the reviewer for their positive comment.
Point 7. The manuscript contains multiple minor grammatical errors or slightly awkward wording which will be readily apparent to native English speakers (e.g., line 40, “begins to opens up…”). Other lines where careful reading will reveal minor grammatical flaws include 146, 204, 251, 266, 280, 292, in lines 262-264.
Response 7: Thank you for pointing this out, we corrected the text following the reviewer’s suggestions.
Point 8. Finally, it is unclear whether the format of the references is that used by MDPI/Diagnostics.
Response 8: We thank the reviewer for this observation. We modified the format of the references.
Round 2
Reviewer 1 Report
Congratulations!
Author Response
We thank the Reviewer for their positive comment and careful review, which helped improve the manuscript.
Reviewer 2 Report
The authors have answered my questions. However, they marked the wrong line numbers. I cannot judge whether they have made the appropriate modification of their manuscript. Please correct it.
Round 3
Reviewer 2 Report
The line number is still wrong in the reply letter. I only got the PDF file. I don't have the "Word" file. It is the authors' responsibility to make it clear.